# Molecular and Functional Analyses of Characterized Sesquiterpene Synthases in Mushroom-Forming Fungi

**DOI:** 10.3390/jof9101017

**Published:** 2023-10-14

**Authors:** Shengli Wang, Ruiqi Chen, Lin Yuan, Chenyang Zhang, Dongmei Liang, Jianjun Qiao

**Affiliations:** 1Department of Pharmaceutical Engineering, School of Chemical Engineering and Technology, Tianjin University, Tianjin 300072, China; wangshengli_2020@tju.edu.cn (S.W.); chenruiqi0803@163.com (R.C.); yuanlinspxy@163.com (L.Y.); 2Key Laboratory of Systems Bioengineering (Ministry of Education), Tianjin University, Tianjin 300072, China; 3Zhejiang Institute of Tianjin University (Shaoxing), Shaoxing 312300, China; ldmxp@tju.edu.cn; 4Science Center for Future Foods, Jiangnan University, Wuxi 214122, China; zhangchenyang@stu.jiangnan.edu.cn

**Keywords:** sesquiterpenes, sesquiterpene synthases, basidiomycete, fungi, mushroom, *β*-barbatene

## Abstract

Sesquiterpenes are a type of abundant natural product with widespread applications in several industries. They are biosynthesized by sesquiterpene synthases (STSs). As valuable and abundant biological resources, mushroom-forming fungi are rich in new sesquiterpenes and STSs, which remain largely unexploited. In the present study, we collected information on 172 STSs from mushroom-forming fungi with experimentally characterized products from the literature and sorted them to develop a dataset. Furthermore, we analyzed and discussed the phylogenetic tree, catalytic products, and conserved motifs of STSs. Phylogenetic analysis revealed that the STSs were clustered into four clades. Furthermore, their cyclization reaction mechanism was divided into four corresponding categories. This database was used to predict 12 putative STS genes from the edible fungi *Flammulina velutipes*. Finally, three *FvSTSs* were selected to experimentally characterize their functions. *FvSTS03* predominantly produced Δ-cadinol and *FvSTS08* synthesized *β*-barbatene as the main product; these findings were consistent with those of the functional prediction analysis. A product titer of 78.8 mg/L *β*-barbatene was achieved in *Saccharomyces cerevisiae* via metabolic engineering. Our study findings will help screen or design STSs from fungi with specific product profiles as functional elements for applications in synthetic biology.

## 1. Introduction

Sesquiterpenes are a major class of terpenoids. More than 300 types of basic skeletons have been discovered; these skeletons are widely present in plants, fungi, microorganisms, and insects [1]. Sesquiterpene structures present several acyclic, monocyclic, bicyclic, tricyclic, and tetracyclic systems [2]. Owing to their complex structures, inherent bioactivity, and aroma, sesquiterpenes have widespread applications in food [3], pharmaceutical [4], fragrance [5], fuel [6], and agricultural industries [7]. Mushroom-forming fungi are specific fungal groups with the most conspicuous fruiting bodies; for centuries, they have been used as food and traditional medicine [8]. Mushrooms tend to develop several protective strategies to protect the fruiting bodies from organism attack. For example, mushrooms can produce various structurally diverse sesquiterpenes, many of which exhibit antibacterial, antifungal, and cytotoxic activities [9,10]. These sesquiterpenes play a crucial role in inhibiting fungal growth, modifying bacterial motility, and defending against parasites [11,12]. For example, to protect against predators, *Armillaria mellea* can produce toxic protoilludane-type sesquiterpenes [13]. Furthermore, rufuslactone isolated from the fruiting bodies of *Lactarius rufus* exerts antifungal properties against some pathogenic fungi, including *Alternaria alternata* and *Fusarium graminearum* [14]. To date, various sesquiterpenoid natural products with an extensive repertoire of backbone structures have been isolated and characterized from mushrooms [15,16,17]. Elucidating their biosynthetic pathways has garnered considerable attention [18,19,20].

The biosynthesis of sesquiterpene natural products originates from the common precursor farnesyl pyrophosphate (FPP), which is derived from the C5 unit isopentenyl diphosphate and its isomer dimethylallyl diphosphate [21]. Subsequently, sesquiterpene synthases (STSs) catalyze linear FPP to generate sesquiterpene scaffolds, followed by a series of cyclization reactions and rearrangements, resulting in the synthesis of structurally diverse sesquiterpenoids [22]. To date, more than 150 STSs have been cloned, purified, and biochemically characterized from mushroom-forming fungi, including *Postia placenta* [18], *Phanerochaete chrysosporium* [23], *Ganoderma lucidum* [24], and *Lactarius deliciosus* [25]. Furthermore, the site-specific mutations and cyclization mechanisms of a few STSs have been investigated [26,27,28,29,30]. However, owing to these fungi’s complex life cycle and frequently poor growth under laboratory conditions, fungal STSs, particularly those in mushrooms (basidiomycetes), are not well studied compared with those in plants. In general, each basidiomycete contains, on average, more than 12 STS homologs [31]; this indicates that fungal STSs represent rich but largely unexploited natural resources. Over the past decade, continuous advances in sequencing technologies have led to the accumulation of a large amount of fungal genomic data, facilitating genome mining to discover new STSs.

Overall, our study aims to assemble a comprehensive dataset of experimentally characterized mushroom STSs to elucidate the relationship between STS sequences and their catalytic products, then we used the database as a reference to predict and exploit unidentified STSs from other species. Firstly, previously reported 172 STSs were collated and analyzed by phylogenetic, protein domain, and motif analyses. Then, based on the functional prediction of STSs, three new STS genes from *Flammulina velutipes* were experimentally characterized. Finally, we expressed *β*-barbatene in *Saccharomyces cerevisiae* and achieved the highest yield of 78.8 mg/L. On the one hand, this work offers rich functional elements to researchers for conducting synthetic biology research. On the other hand, it provides a reference for exploring new STSs with novel products, superior activity, and selectivity in nature.

## 2. Materials and Methods

### 2.1. Literature Search for Characterized STSs

The collected STSs were found by manually searching the articles published in PubMed that demonstrated the ability of STSs via in vivo or in vitro experimental characterization. The amino acid sequences and corresponding IDs were collected from the National Center for Biotechnology Information (NCBI) (http://www.ncbi.nlm.nih.gov/), UniProtKB (http://www.uniprot.org/), and JGI (http://www.jgi.doe.gov/) collections (date last accessed on 27 July 2023).

### 2.2. Phylogenetic Tree Construction

Clustal W from MEGA7 was used to perform multiple sequence alignments of the proteins of the 172 STSs. Then, MEGA7 was used to develop a phylogenetic tree using the neighbor joining method. A bootstrap of 1000 replicates was performed [32].

### 2.3. Analysis of Protein Motifs and Domains

The specific domains of STS proteins were identified using Pfam (http://pfam-legacy.xfam.org/) (accessed on 15 August 2023). Furthermore, the conserved motifs in the identified STS proteins were predicted using the online tool Multiple Expectation Maximization for Motif (MEME) (https://meme-suite.org/meme/doc/meme.html) (accessed on 15 August 2023) using the default parameters.

### 2.4. Plasmids and Strain Construction

Appendix A and Appendix A present the plasmids and strains used and constructed in this study, respectively. Appendix A lists the primers used in this study. Appendix A lists the codon-optimized gene sequences. Phanta max super-fidelity DNA polymerase (Vazyme, P505, Nanjing, China) was used for PCR amplification. The recombinant strain was constructed using our previously described method [33]. The lithium acetate method was used for yeast transformation [34].

### 2.5. Heterologous Expression of STSs in Yeast

The *p*ESC-URA plasmids containing the codon-optimized *FvSTS* genes were transformed into an engineered *S. cerevisiae* strain Sc027 [35] to produce sesquiterpenes. In addition, an empty *p*ESC-URA vector was heterologously expressed in Sc027 as a control plasmid. The recombinant yeasts were cultured in 10 mL of synthetic complete (SC) drop-out medium (20 g/L glucose, 6.7 g/L yeast nitrogen base, and 2 g/L amino acid drop-out mix) at 220 rpm and 30 °C for 18 h. Subsequently, the culture solution was inoculated into 50 mL of SC medium with an initial OD_600_ of 0.05 and cultured at 30 °C and 200 rpm for 30 h. Then, galactose was added at a final concentration of 10 g/L for inducible protein expression. In situ product extraction into an organic phase is a widely used method to minimize the loss due to evaporation and improve the fermentative production of sesquiterpenes [36]. Thus, 5 mL of dodecane was added to the capture product in the fermentation medium after adding galactose. After culturing for 120 h, the fermentation broth was transferred into a 50 mL tube and centrifuged at 10,000 g for 10 min. The dodecane phase was collected and dehydrated by anhydrous sodium sulfate, then filtered using a 0.22 μm filter and analyzed via gas chromatography–mass spectroscopy (GC–MS).

### 2.6. GC–MS Analysis

The assay products were analyzed using an 8890–7000D GC–MS system equipped with a 7693A automatic liquid sampler and flame ionization detector (Agilent Technologies, Santa Clara, CA, USA). GC analysis was performed on an HP-5MS capillary column (30 m × 0.25 mm × 0.25 µm). One microliter of each dodecane sample was injected into the system at a split ratio of 1:10. Helium was used as the carrier gas at a constant flow rate of 1 mL/min. The temperature of the inlet and detector were set to 280 °C and 200 °C, respectively. The oven temperature was maintained at 70 °C for 2 min and then gradually increased to 300 °C at a rate of 10 °C/min. The MS scan range (m/z) was 35–350 [37]. The fermentation products were identified by comparing their MS spectra and retention times with the NIST17 library. The standard *β*-caryophyllene was dissolved in dodecane and used to construct the standard curves for quantification. 

## 3. Results

### 3.1. Database of Characterized STSs

To obtain a comprehensive dataset of functional STSs in mushrooms, the public databases NCBI, UniProt, and JGI were searched and studies on enzymes with characteristic STS domains in mushrooms were manually reviewed. In total, 174 STSs identified from 35 basidiomycetes have been experimentally characterized in previous studies. Table 1 and Appendix A present the information on these 174 STSs, including gene name, species origin, GenBank or JGI protein ID, major and minor products, and type of cyclization reaction. Among these, the amino acid sequence information of 172 STSs was available (File S1). Apart from these functionally characterized STSs, thousands of putative STSs have been identified in sequenced mushroom genomes and transcriptomes; however, their product specificities remain unknown [17].

To understand the relationship between the STS sequences and their product structures, a protein sequence-based phylogenetic tree of the 172 STSs was constructed using the neighbor joining method with the Jones–Taylor–Thornton model and pairwise deletion with 1000 bootstrap replicates using MEGA7 software. The phylogenetic tree revealed that the STS proteins were clustered into four distinct clades: clades I, II, III, and IV (Figure 1); this finding is consistent with previous studies [16,61,62]. Clustering via sequence conservation suggested that the STSs within one specific clade catalyze the same or a related cyclization reaction. The STSs in clade I produce terpenes such as *α*-muurolene, germacrene A, and cadinoles via the 1,10-cyclization of (*2E*,*6E*)-FPP with the (*E*,*E*)-germacradienyl cation as an intermediate. The STSs in clade II catalyze the 1,10-ring closure of the FPP stereoisomer (*3R*)-nerolidyl diphosphate (*(3R)-*NPP) to produce the intermediate (*Z*,*E*)-germacradienyl cation. The STSs in clade III are characterized by the 1,11-cyclization of (*2E*,*6E*)-FPP via a trans-humulyl cation intermediate. Lastly, the STSs in clade IV initiate 1,6-ring closure using (*3R*)-NPP as a substrate to yield the intermediate (*6R*)-*β*-bisabolol cation (Figure 2) [31,38,42]. 

### 3.2. Phylogenetic Analysis

To better understand the evolutionary relationships of the STSs in each clade, phylogenetic analysis was performed separately (Figure 3). The first clade (clade I) comprised 33 representative STSs from 18 different basidiomycetes, with STSs from *Craterellus cinereus* (Cop1–3) and *Omphalotus olearius* (Omp1 and Omp3), which are known to use the 1,10-cyclization of (*2E*,*6E*)-FPP to produce sesquiterpenes derived from the (*E*,*E*)-germacradienyl cation. Cop1 and Cop2 can synthesize germacrene A as the major product, whereas Cop3, Omp1, and Omp3 can synthesize *α*-muurolene as the main product [31,38]. Most STSs in clade I were identified as promiscuous enzymes that produce multiple sesquiterpene scaffolds. Of these STSs, 12 STSs could synthesize *α*-muurolene as the main product, whereas 8 STSs could synthesize *α*-muurolene as a side product. Furthermore, nine STSs that can produce cadinols as the main product were identified (Figure 3a). The second clade (clade II) comprised 41 STSs from 22 species. Most of the STSs (Cop4, Omp4, Omp5a, Omp5b, etc.) in clade II could transform FPP to generate *β*-copaene or cadinene via its isomer (*3R*)-NPP. Similarly, most STSs found in this clade also generated multiple products. Furthermore, clade II was divided into three clusters, with 17 of the 18 STSs that could produce cadinene as the major product distributed in a larger cluster and 6 STSs that could synthesize viridiflorol as the main product being grouped in the other cluster. Notably, LbSTS4a could produce the acyclic sesquiterpene (*E*)-nerolidol and form a separate cluster within clade II (Figure 3b). 

The third clade (clade III) comprised 50 STSs from 21 different species. Nearly 50% (23) of the STSs grouped in this clade could produce tricyclic sesquiterpenes, including Δ^6^-protoilludene, pentalenene, and aromadendrene, which are derived from the common transhumulyl cation intermediate (Figure 3c). Furthermore, most of the STSs identified in this clade could generate a single major product. For example, AvSTS01 can synthesize a sesquiterpene alcohol as a single major product, whereas PoSTS06 can synthesize a sesquiterpene hydrocarbon as a single major product [56]. Moreover, PpSTS08 can specifically produce Δ^6^-protoilludene [18]. The last clade (clade IV) comprised 48 representative STSs from 22 different mushroom species. The STSs found in this clade could synthesize cadinane-type sesquiterpenoids. Eight STSs appeared in the same small branch and were all specific enzymes coding a single product, *γ*-cadinene (Figure 3d). Interestingly, the STSs grouped in this clade could also synthesize acyclic (*E*)-nerolidol, *α*-farnesene and *β*-farnesene, monocyclic (*Z*)-*α*-bisabolene, bicyclic *α*-cuprenene, tricyclic *α*-santalene, *α*-barbatene, and *β*-barbatene. Therefore, the STSs that appear in this clade may be functionally diverse to synthesize different types of sesquiterpene skeletons. A simple phylogenetic analysis can offer a predictive framework for discovering more STSs from underexploited mushroom-forming fungi in nature.

### 3.3. Analysis of Protein Conserved Motifs

STSs have well-known conserved domains that contain aspartate-rich and NSE/DTE motifs, which play important roles in coordinating Mg^2+^ to facilitate the ionization of FPP/NPP in the active site [63]. First, Pfam analysis was performed to identify the specific protein domains of the 172 STSs. In total, 144 STSs were identified to contain the following conserved domains (Table 2 and Appendix A): PF19086 (Terpene_syn_C_2) or PF03936 (Terpene_synth_C), which correspond to the C-terminal domain of terpene synthase. Twenty-five STSs had the domain with the Pfam ID PF06330 (TRI5), described as trichodiene synthase, which is a terpenoid cyclase domain that catalyzes the FPP cyclization to form the bicyclic sesquiterpene hydrocarbon trichodiene [64]. Two STSs from *Antrodia cinnamomea*, namely Tps1A and Tps2A, were characterized by the UbiA prenyltransferase domain (Pfam ID: PF01040). The other three TPSs, namely, AncA, AncC, and DS3, were characterized by the HAD_2 domain (Pfam ID: PF13419), described as a haloacid dehalogenase-like hydrolase domain. Five STSs (Tps1A, Tps2A, AncA, AncC, and DS3) appeared in the same small branch and formed a separate cluster within clade IV (Figure 3d). Furthermore, the conserved motifs of the 172 STS proteins were analyzed using MEME software. The NDxxSxxxE (NSE) motif was identified as the most conserved motif, covering 161 of the 172 STS protein sequences; in contrast, the asparagine-rich regions [D(D/E/N)xx(D/E)] had various sequences among the 172 STSs. The DExxD sequence was often observed in an appropriate position in 87 STSs. Furthermore, the DD(N)xxD sequence was identified in the aspartate-rich region of 53 STSs. Sequence conservation analysis of these STSs supports more effective site-directed mutagenesis to modulate enzyme activity and specificity and helps us to understand the cyclization mechanisms.

### 3.4. Characterization of FvSTSs

Previous studies have reported that *F. velutipes* is rich in bioactive sesquiterpenes, including sixteen cuparene-type sesquiterpenes, enokipodins with antimicrobial activity, and flammulinolides with antitumor and anticancer activities [65,66,67,68]. To better understand sesquiterpene biosynthesis in *F. velutipes*, candidate STS sequences in the genomic database of *F. velutipes* (ASM1180015v1) were searched using BLAST and the 172 STSs in our database. Twelve putative STS genes were identified, which were named *FvSTS01–12* (Appendix A). The twelve *FvSTSs* were widely distributed in the phylogenetic tree, and four STSs belonged to clade III, six to clade IV, one to clade I, and one to clade II (Figure 4). To verify the results of our bioinformatics analysis, *FvSTS03*, *FvSTS08*, and *FvSTS11* were selected for experimental characterization. The codon optimization is the most critical determinant of increasing heterologous protein expression [69]. Thus, three STSs were codon-optimized for expression in *S. cerevisiae* and synthesized into *p*ESC-URA vectors by GENEWIZ (Suzhou, China). Then, three plasmids were transformed into the engineered *S. cerevisiae* strain Sc027. GC–MS analysis (Figure 5a) revealed that *FvSTS03* abundantly produced Δ-cadinol, with small amounts of minor products, including *γ*-muurolene, *α*-muurolene, *β*-cadinene, and *α*-cadinol; this is consistent with the findings of the functional evolutionary analysis. Furthermore, *FvSTS08* produced *β*-barbatene as the main product and *α*-barbatene, dauca-4(11),8-diene, and *α*-cuprenene as side products (Figure 5b), which were also produced by the other STSs in clade IV of the phylogenetic tree. However, no products were detected in *FvSTS11*-expressing *S. cerevisiae* cultures. These results confirm the prediction capability of our method; however, the method still has some limitations.

### 3.5. Heterologous Production of β-Barbatene in S. cerevisiae

Studies have reported that *β*-barbatene can attract insects for spore dispersal and respond to herbivore infestation [70,71,72]. In this work, we obtained *FvSTS08* by a bioinformatics approach, and then experimentally demonstrated that *FvSTS08* can synthesize *β*-barbatene. To improve the heterologous production of *β*-barbatene in yeast, an effective strategy was developed to increase substrate FPP supply by enhancing the mevalonate (MVA) pathway and inhibiting the branch pathways (Appendix A). In a previous study, Sc027 was engineered to increase FPP supplementation by enhancing MVA. The genes *dpp1* (encoding phosphatidate phosphatase) and *lpp1* (encoding phosphatidate phosphatase) were reported to be responsible for converting FPP to farnesol; their knockout can enhance FPP supplementation [73]. Therefore, the genes *dpp1* and *lpp1* were knocked out in Sc027, generating the strain WSL01 (Appendix A). Furthermore, the plasmid *p*ESC-*FvSTS08* was transformed into strain WSL01, resulting in the strain WSL01-*FvSTS08*. GC−MS analysis revealed that WSL01-*FvSTS08* produced higher levels of *β*-barbatene, with a titer of 78.8 mg/L; this was higher than that achieved using the strain Sc027-*FvSTS08* (43.2 mg/L) (Figure 6a). Barbatene has two isomers: *α*-barbatene and *β*-barbatene. Interestingly, we observed a rearrangement of *β*-barbatene to the better-described *α*-isomer under different strong acid conditions (Figure 6b–d). *α*-Barbatene possesses considerable potential as a high-energy aviation fuel [74]. Overall, our study provides an alternative approach for producing *α*-barbatene.

## 4. Discussion

Mushroom-forming fungi are particularly well known for their ability to synthesize several structurally diverse sesquiterpenoids, many of which are used as lead compounds for new drugs owing to their diverse pharmacological activities, including anticancer, antifungal, and antibiotic effects [15]. A comprehensive understanding of fungal STSs can help elucidate the biosynthetic mechanism of sesquiterpenoids, which has gradually become the focus of attention. In the present study, we collected 174 functionally characterized fungal STSs from previous studies. Phylogenetic analysis was performed and 172 STSs were divided into four distinct clades. Similar studies have reported that basidiomycetous STSs can be divided into four distinct clades (clades I–IV) [16,41,50,61]. However, some other studies have revealed that basidiomycetous STSs can be grouped into five distinct clades (clades I–V) [18,19,31,39]. The STSs in clade V possess significant sequence similarity to those in clade IV and probably prefer to catalyze the 1,6-cyclization of NPP to generate the bisabolyl cation intermediate, which is similar to the cyclization mechanism observed in the STSs in clade IV [31,39,75]. 

In general, the phylogenetic tree-based classification of STS protein sequences is consistent with the classification based on the mechanisms of the cyclization reaction of their products. However, some discrepancies remain; therefore, phylogenetic analysis cannot be an accurate predictor of the product specificities of STSs. First, acyclic sesquiterpenes do not undergo cyclization; however, STSs synthesizing acyclic sesquiterpenes appeared in clades II, III, and IV. This may be because acyclic sesquiterpenes (farnesene and nerolidol) are derived from primary cations via proton loss or a reaction with water molecules in the early stage of FPP conversion, which is shared by the biosynthetic pathways of multiple sesquiterpenes. Second, sesquiterpene biosynthesis may occur via different cyclization reactions. For example, in a previous study, researchers proposed a mechanism for viridiflorol formation based on quantum chemical calculations, starting with the 1,10-cyclization of (*2E*,*6E*)-FPP [76]. Phylogenetic analysis revealed that Agr5 belongs to clade II (Figure 3b); in contrast, SiTPS was placed in Clade III (Figure 3c). This result indicates that viridiflorol biosynthesis can occur via both routes. In addition, (−)-germacrene D can be synthesized from farnesyl cations via both routes: 1,10 or 1,11-cyclization. Although each enzyme may only follow one cyclization route to form (−)-germacrene D, to date, this route remains unelucidated [77]. Third, differences are often observed between the product profiles of STSs encoded by homologous genes from the same or related species in the same clade. For example, PcSTS03 and PcSTS04 from *P. chrysosporium* are the closest on the evolutionary tree (Figure 3a); however, PcSTS03 predominantly produces epicubenol, whereas PcSTS04 synthesizes Δ-cadinene as the main product [23]. This may be because some STSs are promiscuous enzymes that may convert the substrate FPP into various side products via cascade reactions of hydroxylation, elimination, cyclization, and rearrangement [78,79]. Furthermore, a single substitution of an amino acid residue may significantly alter the product profiles of STSs [26,55]. 

Many STSs have been characterized in plants and fungi; however, information on bacterial STSs is scarce. Typical STSs comprise two conserved metal-binding motifs. The first conserved motif is the aspartate-rich region. In a previous study, the canonical form of the aspartate-rich region DDxx(D/E) was identified in 247 of 249 spermatophyte enzymes in plants [77]. However, in our database, the aspartate-rich regions of fungal STSs had various sequences (D(D/E/N)xx(D/E)). The DExxD sequence was often observed in an appropriate position in basidiomycetous STSs. The second conserved motif is called the NSE/DTE motif. The consensus sequence NDxxSxxxE was identified in 161 of the 172 fungal STSs; however, the NSE/DTE motif of plant spermatophyte STSs has various sequences ((N/D)Dxx(S/T/G)xxxE) [57]. In addition to the two conserved motifs, other characteristic conserved motifs, including DxDTT, DDxDTT, and QDxxDxxxD, are present in fungal STS sequences. 

In a previous study, 30 sesquiterpenes were isolated from the solid and liquid cultures of *F. velutipes*, including *β*-cadinene and *α*-muurolene [46]. Furthermore, six oxygenated cuprenene derivatives were isolated from a solid culture of *F. velutipes* growing on cooked rice [80]. In our study, we found that *FvSTS03* of *F. velutipes* can produce small amounts of *α*-muurolene and *β*-cadinene and that *FvSTS08* of *F. velutipes* can produce cuprenene as a minor product. Other products such as *β*-barbatene and Δ-cadinol identified in this study have not been discovered from *F. velutipes*. This may be because the contents of these products are very low in vivo or because these products are only released under specific stress conditions.

## 5. Conclusions

In the present study, we collected the information of mushroom STSs with experimentally identified functions and constructed a phylogenetic tree of mushroom functional STSs based on the amino acid sequences. The catalytic products, and conserved domains and motifs of these STSs were analyzed and discussed to explore the sequence–structure–function relationships. Then, our database was applied to predict 12 putative *FvSTS* genes from *F. velutipes* and 3 *FvSTS* genes were experimentally verified. Finally, the product titer of 78.8 mg/L *β*-barbatene in *S. cerevisiae* was achieved through expressing *FvSTS08*. The approach can also be valuable for exploring other new STSs and sesquiterpenes from mushroom-forming fungi in nature.

## Figures and Tables

**Figure 1 jof-09-01017-f001:**
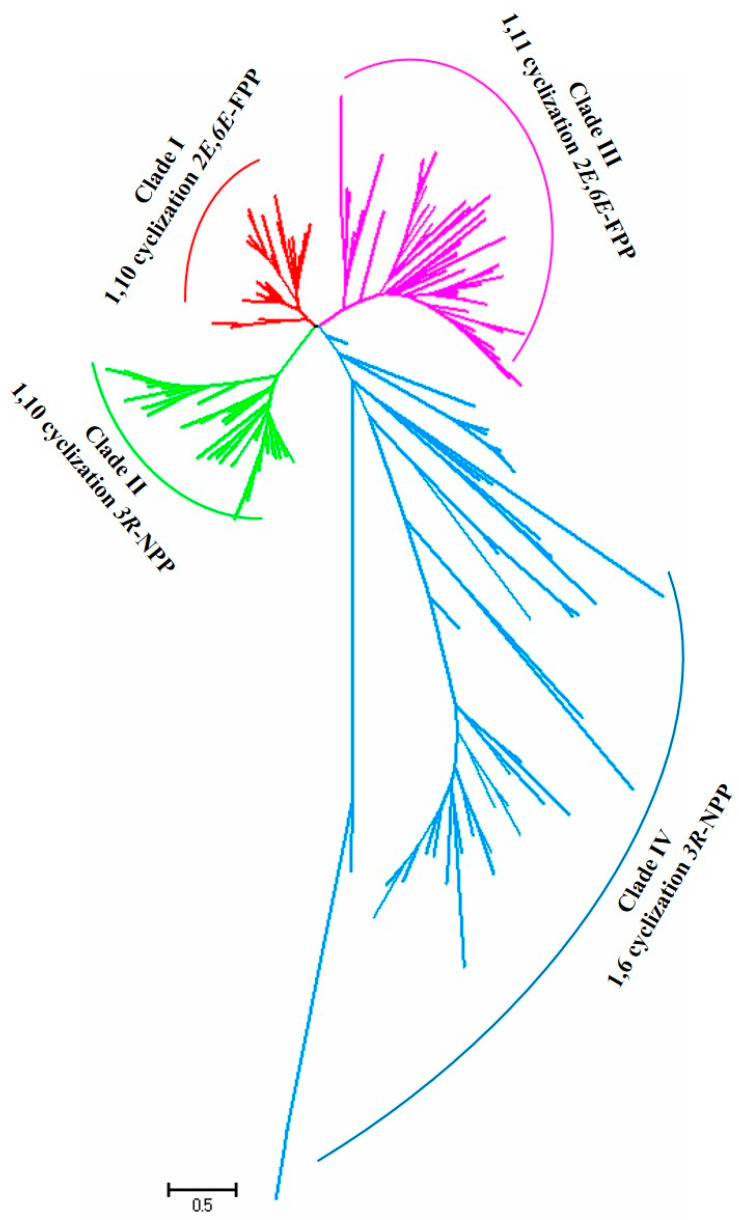
Phylogenetic analysis of the functional STSs in mushroom-forming fungi. Red lines represent clade I STSs, green lines represent clade II STSs, purple lines represent clade III STSs, and blue lines represent clade IV STSs.

**Figure 2 jof-09-01017-f002:**
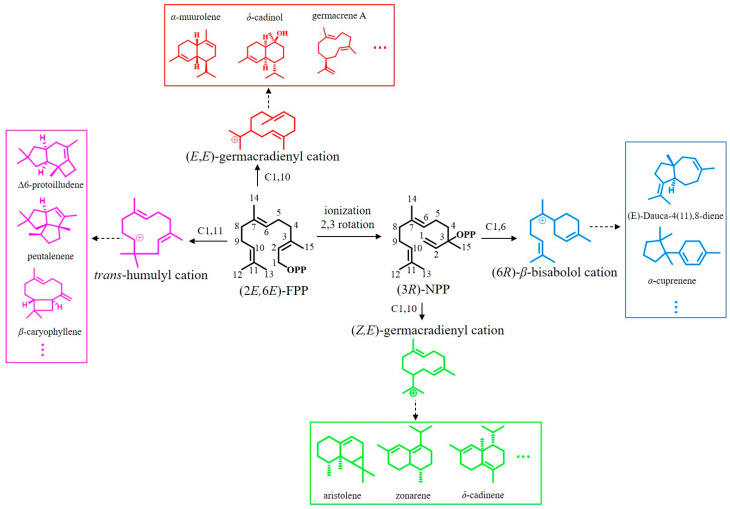
Reaction mechanisms underlying sesquiterpene production starting from FPP in basidiomycetes.

**Figure 3 jof-09-01017-f003:**
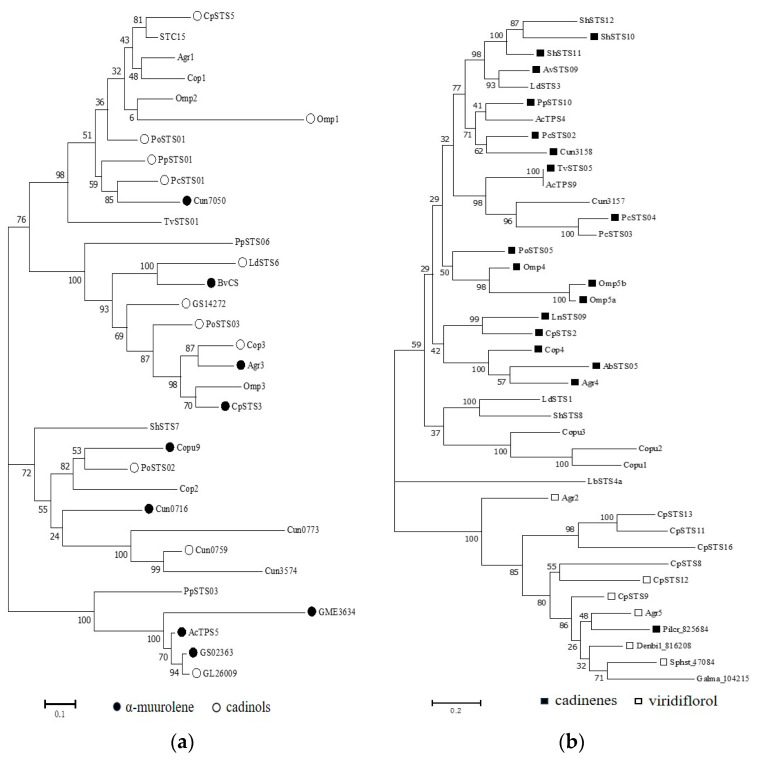
The phylogeny of STSs in each clade. (**a**) Clade I; (**b**) Clade II; (**c**) Clade III; (**d**) Clade IV.

**Figure 4 jof-09-01017-f004:**
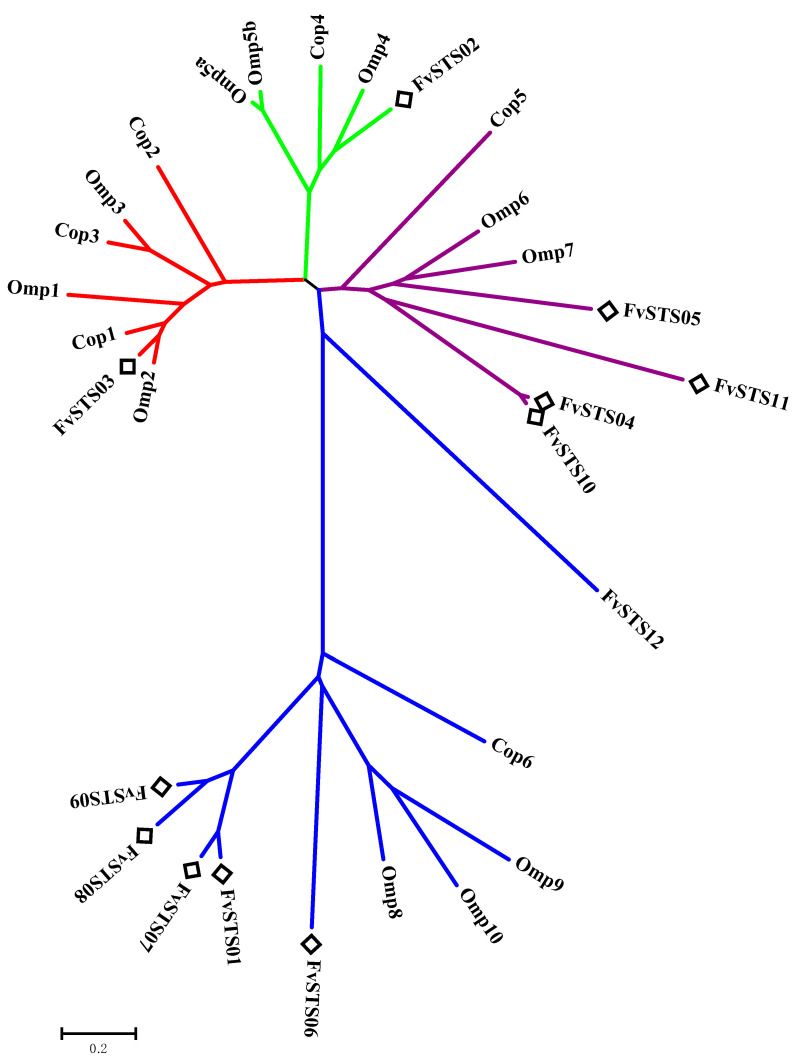
The phylogeny of candidate STSs in *F. velutipes*.

**Figure 5 jof-09-01017-f005:**
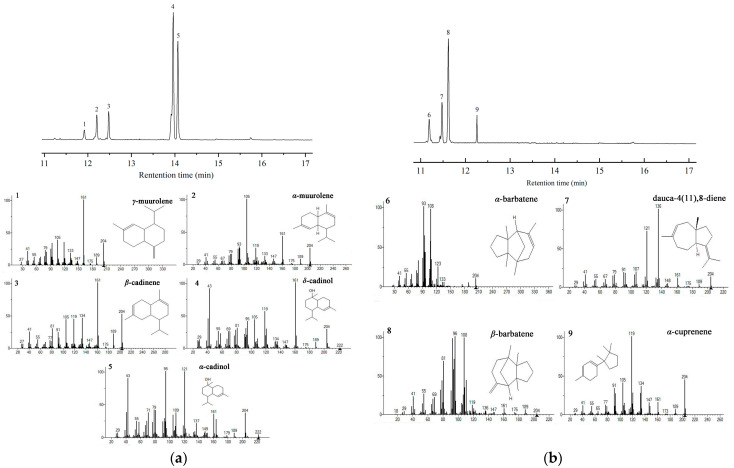
GC–MS analysis of the assay products of *FvSTSs* in engineered yeast: (**a**) *FvSTS03* and (**b**) *FvSTS08*. The produced sesquiterpenes were putatively annotated as 1, *γ*-muurolene; 2, *α*-muurolene; 3, *β*-cadinene; 4, Δ-cadinol; 5, *α*-cadinol; 6, *α*-barbatene; 7, dauca-4(11),8-diene; 8, *β*-barbatene; and 9, *α*-cuprenene.

**Figure 6 jof-09-01017-f006:**
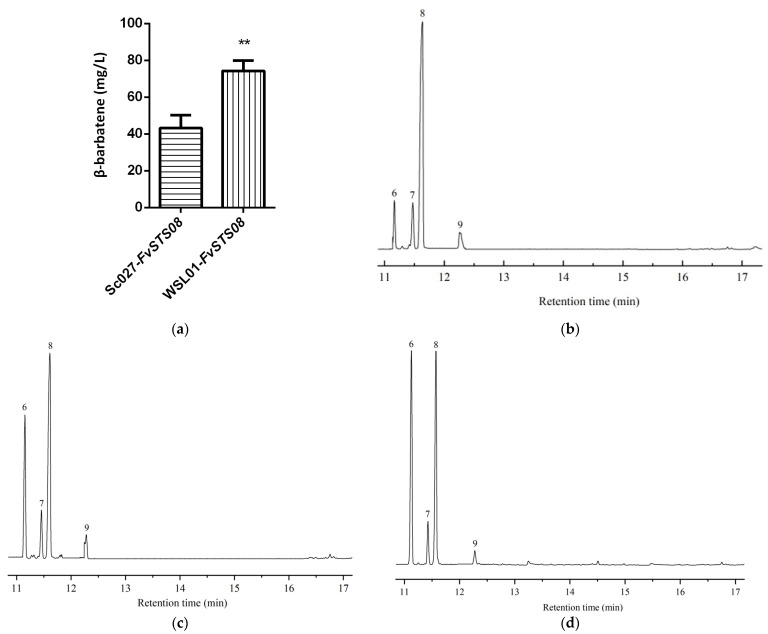
Heterologous production of *β*-barbatene in *S. cerevisiae* and gas chromatograms of the WSL01-*FvSTS08* strain under different acid treatment conditions. (**a**) *β*-Barbatene production; (**b**) 3 M HCl treatment; (**c**) 3 M H_2_SO_4_ treatment; and (**d**) 3 M HNO_3_ treatment. The produced sesquiterpenes were putatively annotated as 6, *α*-barbatene; 7, dauca-4(11),8-diene; 8, *β*-barbatene; and 9, *α*-cuprenene. The value is the mean of three independent experiments. (** represents *p* < 0.01).

**Table 1 jof-09-01017-t001:** Sesquiterpene synthases cloned in mushroom-forming fungi to date.

Species	Gene Name	GenBank or JGI Protein ID	Products	Clade	Reference
*Lactarius* *deliciosus*	LdSTS1	KAH9077233.1	1,10-di-epi-cubenol	II	[25]
	LdSTS2	KAH9033224.1	ND	III	
	LdSTS3	KAH9035467.1	myrcene, trans-*β*-ocimene, etc.	II	
	LdSTS6	KAH9079282.1	*α*-muurolene, *γ*-gurjunene, *α*-selinene, etc.	I	
	LdSTS7	KAH9033225.1	aristolene	III	
	LdSTS8	KAH9071064.1	ND	III	
	LdSTS10	KAH9071064.1	ND	III	
	LdSTS11	KAH9053987.1	aristolene	III	
	LdSTS12	KAH9033225.1	ND	III	
	LdSTS14	KAH9033227.1	ND	III	
*Coprinus* *cinereus*	Cop1	XP_001832573.1	germacrene A; Δ-cadinene; *α*-muurolene; germacrene D	I	[38]
	Cop2	XP_001836556.1	germacrene A; Δ-cadinene; *α*-muurolene	I	
	Cop3	XP_001832925.1	*α*-muurolene; germacrene A; *γ*-muurolene; germacrene D; Δ-cadinene; *α*-copaene	I	
	Cop4	XP_001836356.1	Δ-cadinene; *β*-copaene; *β*-cubebene; sativene; germacrene D; cubebol	II	
	Cop5	XP_001834007.1	ND	III	
	Cop6	XP_001832549.1	*α*-cuprenene	IV	
*Omphalotus* *olearius*	Omp1	JGI ID: 1311	*α*-muurolene	I	[31]
	Omp2	not available	ND	I	
	Omp3	JGI ID: 4636	germacrene A; *α*-muurolene; elina-4,7-diene; Δ-cadinene	I	
	Omp4	JGI ID: 1447	Δ-cadinene	II	
	Omp5a	JGI ID: 2392	*γ*-cadinene; epi-zonarene; germacrene A	II	
	Omp5b	JGI ID: 2393	*γ*-cadinene; germacrene A	II	
	Omp6	JGI ID: 4774	Δ^6^-protoilludene	III	
	Omp7	JGI ID: 2271	Δ^6^-protoilludene; pentalenene	III	
	Omp8	not available	ND	IV	
	Omp9	JGI ID: 3258	*α*-barbatene; *β*-barbatene	IV	
	Omp10	JGI ID: 3981	(*E*)-dauca-4(11),8-diene; daucene	IV	
*Stereum* *hirsutum*	ShSTS1	JGI ID: 159379	*β*-barbatene; *α*-barbatene	IV	[39]
	ShSTS11	JGI ID: 128017	Δ-cadinene; *α*-cubebene	II	
	ShSTS18	JGI ID: 25180	Δ^6^-protoilludene	III	
	ShSTS15	JGI ID: 64702	Δ^6^-protoilludene	III	
	ShSTS16	JGI ID: 73029	Δ^6^-protoilludene	III	
	ShSTS3	JGI ID: 122776	*α*-farnesene; *β*-farnesene	IV	
	ShSTS4	JGI ID: 52743	hirsutene	IV	
	ShSTS5	JGI ID: 161672	*γ*-cadinene	IV	
	ShSTS7	JGI ID: 167646	Δ-cadinene	I	
	ShSTS8	JGI ID: 146390	1-epi-cubenol; *α*-cubebene	II	
	ShSTS10	JGI ID: 111121	Δ-cadinene; germacrene D	II	
	ShSTS12	JGI ID: 111127	*α*-cubebene; *β*-cubebene	II	
	ShSTS13	JGI ID: 50042	*β*-caryophyllene	III	
	ShSTS17	JGI ID: 69906	Δ^6^-protoilludene	III	
	HS-HMGS	not available	hirsutene; *β*-caryophyllene	IV	[40]
*Clitopillus pseudo-pinsitus*	CpSTS1	BBH51498.1	sterpurene	III	[41]
	CpSTS2	BBH51499.1	Δ-cadinene; *α*-cubebene	II	
	CpSTS3	BBH51500.1	Δ-cadinol; *α*-muurolene; *γ*-muurolene; unknown sesquiterpene	I	
	CpSTS4	BBH51501.1	Δ^6^-protoilludene	III	
	CpSTS5	BBH51502.1	*α*-muurolene; *γ*-muurolene	I	
	CpSTS6	BBH51503.1	pentalenene	III	
	CpSTS7	BBH51504.1	*α*-farnesene	III	
	CpSTS8	BBH51505.1	alloaromadendrene; unknown sesquiterpene	II	
	CpSTS9	BBH51506.1	virifloridol; ledene	II	
	CpSTS11	BBH51508.1	9-alloaromadendrene	II	
	CpSTS12	BBH51509.1	virifloridol; *β*-elemene; ledene	II	
	CpSTS13	BBH51510.1	ledene; unknown sesquiterpene	II	
	CpSTS14	BBH51511.1	*β*-elemene; *β*-farnesene; *α*-farnesene	IV	
	CpSTS15	BBH51512.1	ND	IV	
	CpSTS16	BBH51513.1	aristolene; unknown sesquiterpene	II	
	CpSTS17	BBH51514.1	*β*-caryophyllene	IV	
	CpSTS18	BBH51515.1	*γ*-cadinene	IV	
*Antrodia* *cinnamomea*	AcTPS4	JGI ID: 40411	zonarene; *α*-cubebene; sibirene; *γ*-cadinene	II	[42]
	AcTPS5	JGI ID: 40579	*T*-cadinol; *γ*-cadinene	I	
	AcTPS7	JGI ID: 36944	nerolidol; *α*-farnesol	\	
	AcTPS9	JGI ID: 47706	1-epi-cubenol; sibirene; cubebol; *α*-cubebene; *α*-farnesol; *γ*-muurolene	II	
	Tps1A	KAI0942648.1	(+)-(*S*,*Z*)-*α*-bisabolene	IV	[43]
	Tps2A	KAI0928020.1			
	AncA	ACg006372	(*R*)-trans-*γ*-monocyclofarnesol	IV	[44]
	AncC	ACg006375	drimane-type sesquiterpene (+)-albicanol		
*Boreostereum vibrans*	BvCS	KU668561.1	Δ-cadinol; *α*-muurolene; *γ*-muurolene	I	[45]
*Lignosus* *rhinocerotis*	GME3634	KX281943	*α*-cadinol; germacrene D-4-ol	I	[46]
	GME3638	KX281944	torreyol; germacrene D-4-ol; *β*-cubebene	IV	
	GME9210	KX281945	1,3,4,5,6,7-hexahydro-2,5,5-trimethyl-2H-2,4a-ethanonaphthalene	III	
*Agrocybe* *aegerita*	Agr1	MN146024	Δ-cadinene; *α*-cadinol; Δ-cadinol; *α*-muurolene	I	[17]
	Agr2	MN146025	viridiflorene	II	
	Agr3	MN146026	Δ-cadinol; Δ-cadinene; *α*-muurolene; *γ*-muurolene	I	
	Agr4	MN146027	Δ-cadinene; epicubenol; cadina-1(6),4-diene; *β*- myrcene	II	
	Agr5	MN146028	viridiflorol; viridiflorene	II	
	Agr6	MN146029	Δ^6^-protoilludene	III	
	Agr7	MN146030	Δ^6^-protoilludene	III	
	Agr8	MN146031	*γ*-muurolene; *β*-cadinene; Δ-cadinol	III	
	Agr9	MN146032	*γ*-muurolene; Δ-cadinene; unknown sesquiterpenol	III	
	Agr10	MN146033	ND	III	
	Agr11	MN146034	ND	III	
*Coniophora* *puteana*	Copu1	XP_007772164.1	ND	II	[47]
	Copu2	XP_007771895.1	*β*-copaene; germacrene D; cubebol; germacrene D-4-ol	II	
	Copu3	XP_007765978.1	cubebol; germacrene D-4-ol; Δ-cadinene	II	
	Copu5	XP_007765330.1	Δ-cadinol; Δ-cadinene; cubebol; *α*-cadinol	IV	[30]
	Copu9	XP_007765560.1		I	
*Piloderma* *croceum*	Pilcr_825684	JGI ID: 825684	*γ*-cadinene; viridiflorene; *β*-elemene	II	[17]
*Galerina* *marginata*	Galma_104215	JGI ID: 104215	*β*-gurjunene	II	
*Sphaerobolus stellatus*	Sphst_47084	JGI ID: 47084	viridiflorol; viridiflorene	II	
*Dendrothele bispora*	Denbi1_816208	JGI ID: 816208	viridiflorol; viridiflorene	II	
	Denbi1_659367	JGI ID: 659367	Δ^6^-protoilludene	III	
*Heterobasidion annosum*	HEtan2_454193	XP_009550163.1	Δ^6^-protoilludene	III	
*Hypholoma sublateritium*	Hypsu1_138665	A0A0D2L718.1	Δ^6^-protoilludene	III	
*Armillaria* *gallica*	Pro1	MT277003.1	Δ^6^-protoilludene	III	[48]
*Hypholoma* *fasciculare*	Hfas94a	MK287936.1	*α*-humulene; *β*-caryophyllene	III	[49]
	Hfas94b	MK287937.1	*α*-humulene; *β*-caryophyllene	IV	
	Hfas255	not available	ND	\	
	Hfas344	MK287938.1	unknown sesquiterpene	III	
*Cerrena* *unicolor*	Cun3817	JGI ID: 3817	*γ*-cadinene	IV	[50]
	Cun5155	JGI ID: 5155	aromadendrene	III	
	Cun3157	JGI ID: 3157	*β*-cubebene; germacrene D; epicubenol; Δ-cadinene	II	
	Cun3158	JGI ID: 3158	Δ-cadinene; germacrene D; *β*-cubebene; *γ*-amorphene	II	
	Cun0773	JGI ID: 0773	germacrene D	I	
	Cun7050	JGI ID: 7050	Δ-cadinol	I	
	Cun0716	JGI ID: 0716	Δ-cadinol; *α*-muurolene	I	
	Cun0759	JGI ID: 0759	*α*-muurolene	I	
	Cun3574	JGI ID: 3574	*α*-copaene	I	
	Cun9106	JGI ID: 9106	unknown sesquiterpene	IV	
*Postia* *placenta*	PpSTS01	XP_024337827.1	*α*-muurolene; Δ-cadinene; *β*-elemene	I	[18]
	PpSTS03	A0A348B781.1	*γ*-cadinene; *α*-cadinene; Δ-cadinene; *β*-elemene	I	
	PpSTS06	A0A348B782.1	*α*-gurjunene; bicycloelemene; bicyclogermacrene	I	
	PpSTS08	A0A348B784.1	Δ^6^-protoilludene	III	
	PpSTS09	A0A348B785.1	unknown sesquiterpene	III	
	PpSTS10	XP_024334632.1	Δ-cadinene; *β*-copaene; sativene; sesquisabinene	II	
	PpSTS14	A0A348B788.1	pentalenene; caryophyllene	III	
	PpSTS29	A0A348B794.1	unknown sesquiterpene	IV	
*Fomitopsis pinicola*	Fompi1	JGI ID: 84944	*α*-cuprenene	IV	[31]
*Phanerodontia chrysosporium*	PcSTS01	BCX55496.1	*α*-muurolene; Δ-cadinene; *γ*-muurolene; *α*-muurolol	I	[23]
	PcSTS02	BCX55497.1	Δ-cadinene; *β*-copaene; *β*-farnesene; cadina-1(6),4-diene	II	
	PcSTS03	BCX55498.1	epicubenol	II	
	PcSTS04	BCX55499.1	Δ-cadinene; *β*-farnesene; *β*-copaene; epicubenol	II	
	PcSTS06	BCX55500.1	*β*-barbatene; *α*-barbatene	IV	
	PcSTS08	BCX55502.1	(*E*)-*α*-bisabolene	IV	
	PcSTS11	BCX55504.1	*α*-santalene	IV	
*Steccherinum ochraceum*	A8411	not available	hirsutene	IV	[19]
*Ganoderma sinensis*	GsSTS43	PIL26225	*γ*-cadinene	IV	[24]
	GsSTS45a	UDP19925	ND	IV	
	GsSTS45b	UDP19925	*γ*-cadinene	IV	
	GsSTS26	MT584777.1	gleenol; di-epi-1,10-cubenol; Ʈ-muurolol	III	[51]
	GsSTS27	OP094045		III	
	GS02363	PIL35634	*α*-cadinol; Δ-cadinene; *γ*-cadinene; *T*-cadinol	I	[52]
	GS14272	PIL24516	*α*-muurolene	I	[53]
	GS11330	not available	*α*-cuprenene	IV	
*Ganoderma* *lucidum*	GL26009	not available	*α*-muurolene; *γ*-muurolene	I	[54]
	GLSTS6	UDP19923	*γ*-cadinene	IV	[24]
*Termitomyces* sp. J132	STC4	KNZ72568.1	(+)-intermedeol; *α*-selinene; *β*-selinene	IV	[55]
	STC9	KAG5341349	*γ*-cadinene	IV	
	STC15	KNZ74377.1	(+)-germacrene D-4-ol; *γ*-cadinene; Δ-cadinene; *α*-cadinene; *β*-elemene	I	
*Agaricus* *bisporus*	AbSTS05	LC712879	cadina-1,4-diene; cadina-1(6),4-diene; Δ-cadinene; zonarene; epicubenol; cadin-4-en-10-ol	II	[56]
	AbSTS07	LC712880	Δ-cadinene; epizonarene	IV	
	AbSTS09	LC712881	(*Z*)-*α*-bisabolene	IV	
*Auriscalpium vulgare*	AvSTS01	LC712882	unknown sesquiterpene	III	
	AvSTS03	LC712883	Δ^6^-protoilludene	III	
	AvSTS06	LC712885	(*E*)-nerolidol	III	
	AvSTS07	LC712886	(*E*)-nerolidol	III	
	AvSTS09	LC712887	cadina-1,4-diene; cadina-1(6),4-diene; Δ-cadinene; *β*-copaene; zonarene; epicubenol; cadin-4-en-10-ol	II	
*Lepista nuda*	LnSTS01	LC712891	Δ^6^-protoilludene	III	
	LnSTS02	LC712892	Δ^6^-protoilludene	III	
	LnSTS04	LC719126	pleostene; isobazzanene	III	
	LnSTS09	LC712895	cadina-1,4-diene; cadina-1(6),4-diene; Δ-cadinene; *β*-copaene; zonarene; epicubenol; cadin-4-en-10-ol	II	
	LnSTS19	LC712898	(*E*)-nerolidol	IV	
	LnSTS20	LC712899	*β*-barbatene	IV	
	LnSTS25	LC712901	unknown sesquiterpene	IV	
	LnSTS27	LC712902	acora-3(7),14-diene	IV	
*Pleurotus* *ostreatus*	PoSTS01	LC712903	*α*-muurolene; isobazzanene; Δ-cadinene; *α*-muurolol	I	
	PoSTS02	LC712904	*α*-muurolene; Δ-cadinene; *α*-muurolol; zonarene	I	
	PoSTS03	LC712905	*α*-muurolene; Δ-cadinene; isobazzanene; *α*-muurolol	I	
	PoSTS05	LC712906	cadina-1,4-diene; cadina-1(6),4-diene; Δ-cadinene; *β*-farnesene; zonarene; epicubenol; cadin-4-en-10-ol	II	
	PoSTS06	LC712907	pleostene	III	
	PoSTS11	LC712908	(*E*)-nerolidol	IV	
	PoSTS16	LC712909	*α*-cuprenene	IV	
*Trametes* *versicolor*	TvSTS01	LC712910	Δ-cadinene; cadin-4-en-10-ol; Ʈ-muurolol	I	
	TvSTS05	LC712912	cadina-1,4-diene; cadina-1(6),4-diene; Δ-cadinene; *β*-copaene; azoarene; epicubenol; cadin-4-en-10-ol	II	
	TvSTS06	LC712913	cadina-1,4-diene; cadina-1(6),4-diene; Δ-cadinene; *β*-copaene; *β*-farnesene; zonarene; epicubenol	III	
	TvSTS07	LC712914	Δ^6^-protoilludene	III	
	TvSTS12	LC712917	*γ*-cadinene	IV	
	TvSTS14	LC712919	*β*-barbatene; *α*-barbatene	IV	
	TvSTS16	LC712920	dauca-4(11),8-diene; isobazzanene	IV	
*Irpex lacteus*	IIIS	JGI ID: Il4946	iltremulanol A	III	[57]
*Serendipita* *indica*	SiTPS	JGI ID: 77541	viridiflorol	III	[58]
*Termitomyces* sp. T153	DS3	not available	unknown sesquiterpene	IV	[59]
*Laccaria* *bicolor*	LbSTS4a	XP_001887869.1	(*E*)-nerolidol	II	[60]
	LbSTS6	XP_001885710.1	*α*-cuprenene; *α*-cuparene	IV	

ND: no product detected; “\”: not analyzed in this study.

**Table 2 jof-09-01017-t002:** Conserved motifs and domains of STSs.

Sequence Name	Motif I D(D/E/N)xx(D/E)	Motif II (NDxxSxxxE)	Domains (Pfam ID)
LdSTS1	DELSD	NDLYSYNME	PF19086
LdSTS2	DEYTD	**Q**DLYSYNNE	PF19086
LdSTS3	DEVSD	NDVYSYNME	PF19086
LdSTS6	DNVSD	NDIFSYNVE	PF19086; PF03936; PF06330
LdSTS7	DEYTD	NDLYSYNIE	PF19086
LdSTS8	DEFSD	NDIASYNVE	PF19086
LdSTS10	DEFSE	NDIASYNVE	PF19086
LdSTS11	DEYTD	NDLYSYNIE	PF19086
LdSTS12	DEYTD	NDLYSYNVE	PF19086
LdSTS14	DEFTD	NDMYSYNIE	PF19086
Cop1	DNLSD	NDIFSFNVE	PF19086
Cop2	DDWLD	NDIFSFNRE	PF19086
Cop3	DNISD	NDIFSYNVE	PF19086; PF03936; PF06330; PF19035
Cop4	DEISD	NDVYSYDME	PF19086; PF03936
Cop5	D**Y**FFD	NDAYSWNVE	PF19086; PF03936
Cop6	DDAF**Q**	NDLLSFYKE	PF06330
Omp1	DNLTD	NDIYSFNIE	PF19086
Omp2	DNLSD	NDIFSYNVE	PF19086
Omp3	DEVSD	NDIFSYNVE	PF19086
Omp4	DEVSD	NDVYSYNKE	PF19086
Omp5a	DELSD	NDVYSYNVE	PF19086
Omp5b	DEVSD	NDVYSYNVE	PF19086
Omp6	DEYSD	NDLCSYNVE	PF19086
Omp7	DEYSD	NDTASYNYE	PF19086
Omp8	DDVFE	NDIMSFYKE	PF06330
Omp9	DDVFE	NDVLSFYKE	PF06330
Omp10	DDIF**P**	NDVLSFYKE	PF06330
ShSTS1	DDSLE	NDLMSFYKE	PF06330
ShSTS11	DEISD	NDVYSYNVE	PF19086
ShSTS18	DEYSD	**Q**DICSYNVE	PF19086
ShSTS15	DEHSD	NDIVSYNIE	PF19086
ShSTS16	DEYSD	NDIVSYNLE	PF19086
ShSTS3	DDWVD	NEASSYVKE	PF19086
ShSTS4	DDYID	NDFFSYLKE	PF19086
ShSTS5	DDLSD	NDLCSFNKE	PF19086
ShSTS7	DDWTD	NDIFSYNVE	PF19086
ShSTS8	DEISD	NDIYSYDME	PF19086
ShSTS10	DEISD	NDVYSYKVE	PF19086
ShSTS12	DEISD	**Q**DVYSYSME	PF19086
ShSTS13	DDILD	NDTFSYRRE	PF19086
ShSTS17	DEHSD	**S**DIVSWNLE	PF19086
HS-HMGS	DDYID	NDFFSYLKE	PF19086; PF01154
CpSTS1	DEYTD	NDMCSYKKE	PF19086
CpSTS2	DELSD	NDVYSYDME	PF19086
CpSTS3	DNISD	NDIFSYNVE	PF19086
CpSTS4	DEYTD	NDLCSFRNE	PF19086
CpSTS5	DNLSD	NDIFSYNVE	PF19086
CpSTS6	DEYSD	NDLYSYNVE	PF19086
CpSTS7	DEITE	NDVFSFKVE	PF19086
CpSTS8	DEYTD	NDVYSYNME	PF19086
CpSTS9	DEYTD	NDLFSYNME	PF19086
CpSTS11	DEATD	NDIHSYNME	PF19086
CpSTS12	DEYTD	NDLYSYNME	PF19086
CpSTS13	DETTD	NDIQSYNME	PF19086
CpSTS14	DDYI**L**	NDIYSYKVE	PF19086
CpSTS15	DDLME	NDLFSYRKE	PF19086
CpSTS16	DESSD	NDIHSYNME	PF19086
CpSTS17	DDIIE	NDLFSYRVE	PF19086
CpSTS18	DDLSD	NDLCSFNKE	PF19086; PF03936; PF06330;
AcTPS4	DEVSD	NDVYSYNME	PF19086
AcTPS5	DDWTD	NDVLSYNAE	PF19086; PF03936
AcTPS9	DEISD	NDLYSYNME	PF19086
Tps1A	D**I**EGD	QDFPDIEFD	PF01040
Tps2A	D**V**AGD	QDFPDIEFD	PF01040
AncA	DDRIE	DDFTDD	PF13419
AncC	DDKIE	DDFTDD	PF13419
BvCS	DNISD	NDVFSYNVE	PF19086; PF03936
GME3634	DDWTD	NDVLSYNAE	PF19086
GME3638	DDWSD	NDLFSYNVE	na
GME9210	DEYSD	NDIVSYNVE	PF19086
Agr1	DNLSD	NDIFSYSVE	PF19086; PF03936
Agr2	DEVTD	NDLYSYNME	PF19086
Agr3	DNISD	NDIFSYNVE	PF19086; PF03936
Agr4	DEVSD	NDVYSYDME	PF19086
Agr5	DEYTD	NDLVSYNME	PF19086
Agr6	DEHTD	NDLCSYNVE	PF19086
Agr7	DEWSD	NDLCSYNVE	PF19086
Agr8	DEYTD	NDMHSYVRE	PF19086; PF03936
Agr9	DEYTD	NDIDSYAME	PF19086
Agr10	DECAD	na	na
Agr11	DEYTD	na	PF19086
Copu1	DELTD	NDVYSYNME	PF19086
Copu2	DDLTD	NDVFSYNRE	PF19086
Copu3	DELSD	NDVYSYNME	PF19086
Copu5	DDWSD	NDVFSYNKE	PF19086; PF03936
Copu9	DDWLD	NDIFSYNKE	PF19086
Pilcr_825684	DELTD	NDLFSYNRE	PF19086
Galma_104215	DEFTD	NDLFSYDME	PF19086
Sphst_47084	DEYTD	NDLFSYNS	PF19086
Denbi1_816208	DEFTD	NDLFSYNME	PF19086
Denbi1_659367	DEHSD	NDLCSYNVE	PF19086
Hetan2_454193	DEYSD	NDIASYNLE	PF19086
Hypsu1_138665	DEHTD	NDLCSYKVE	PF19086; PF03936
Pro1	DEYSD	NDVVSYNLE	PF19086; PF03936
Hfas94a	DEYTD	NDMHSYGLE	PF19086
Hfas94b	DEDLD	NDLISYTKE	PF19086
Hfas344	DEYTD	NDMHSYALE	PF19086
Cun3817	DDLSD	NDLCSFNKE	PF19086; PF03936
Cun5155	DEHSD	NDLFSYNVE	PF19086
Cun3157	DEISD	NDIYSYNME	PF19086
Cun3158	DEVSD	NDVYSYNME	PF19086
Cun0773	DDWSD	NDILSYSKE	PF19086
Cun7050	DNISD	NDIFSYNVE	PF19086; PF03936
Cun0716	DDWSD	NDIFSFNVE	PF19086
Cun0759	DDWSD	NDIFSYNKE	PF19086
Cun3574	DDWTD	NDIFSYNKE	PF19086
Cun9106	**N**DDYE	na	PF06148
PpSTS01	DNISD	NDIFSYNVE	PF19086
PpSTS03	DDWSD	NDILSYNRE	PF19086; PF03936
PpSTS06	DDITD	NDIYSFNNE	PF19086
PpSTS08	DEYTD	NDLVSYNRE	PF19086
PpSTS09	DEYSD	NDMLSWNVE	PF19086
PpSTS10	DEVSD	NDVYSYNME	PF19086
PpSTS14	DEYTD	NDIASYNKE	PF19086
PpSTS29	DEPD**I**	NDILSFYKE	PF06330
Fompi1	DDPD**I**	NDILSFYKE	PF06330
PcSTS01	DNISD	NDIFSYNVE	PF19086; PF03936
PcSTS02	DEVSD	NDVYSYKME	PF19086
PcSTS03	DEISD	NDVYSYDME	PF19086
PcSTS04	DEISD	NDVYSYDME	PF19086
PcSTS06	DDFE**I**	NDLLSFYKE	PF06330
PcSTS08	DDEA**I**	NDILSFYKE	PF06330
PcSTS11	DDCE**I**	NDIYSFHKE	PF06330
A8411	DDYID	NDLFSYAKE	PF19086
GsSTS43	DDLSD	NDLCSFNKE	PF19086; PF03936
GsSTS45a	DDLSD	NDLCSFNKE	PF19086
GsSTS45b	DDLSD	NDLCSFNKE	PF19086
GsSTS26	DEYTD	NDVASYNRE	PF19086
GsSTS27	DEYTD	NDVASYNRE	PF19086
GS02363	DDWTD	NDVLSYNAE	PF19086; PF03936
GS14272	DNISD	NDIFSYNVE	PF19086
GS11330	DDLGE	NDILSFYKE	PF06330
GL26009	DDWTD	NDVLSYNAE	PF19086; PF03936
GLSTS6	DDLSD	NDLCSFNKE	PF19086; PF03936
STC4	D**R**LTD	NDLYSYKKE	PF19086
STC9	DDLSD	NDLCSFNKE	PF19086; PF03936
STC15	DNLSD	NDIFSYNVE	PF19086; PF03936
AbSTS05	DEISD	NDVYSYNVE	PF19086
AbSTS07	DDNFD	NDITSFYKE	PF06330
AbSTS09	DDNYD	NDIASFYKE	PF06330
AvSTS01	DEYTD	NDLCSYNKE	PF19086; PF03936
AvSTS03	DEYSD	NDIASYNLE	PF19086; PF03936
AvSTS06	DEFTD	NDTYSYNIE	PF19086
AvSTS07	DEFTD	NDTYSYNIE	PF19086
AvSTS09	DEVSD	NDVYSYNME	PF19086
LnSTS01	DEYSD	NDLCSYNVE	PF19086
LnSTS02	DEHSD	NDLCSYNVE	PF19086
LnSTS04	DEYSD	NDVYSYNKE	PF19086; PF03936
LnSTS09	DELSD	NDVYSYDME	PF19086
LnSTS19	DDVD**S**	NDLLSYHKE	PF06330
LnSTS20	DDMS**S**	NDILSFHKE	PF06330
LnSTS25	DDTS**P**	NDLMSFPKE	PF19086; PF06330
LnSTS27	DDKY**F**	NDIMSFYKE	PF06330
PoSTS01	DNLSD	NDIFSYNVE	PF19086; PF03936
PoSTS02	DDWLD	NDLFSYNVE	PF19086; PF03936
PoSTS03	DNISD	NDIFSYNVE	PF19086; PF03936
PoSTS05	DEVSD	NDVYSYNME	PF19086
PoSTS06	DEFSD	NDVYSWNVE	PF19086
PoSTS11	**E**EITE	NDIYSYKKE	PF19086
PoSTS16	DDIS**S**	NDVLSFYKE	PF06330
TvSTS01	DNICD	NDIFSYNVE	PF19086
TvSTS05	DEISD	NDLYSYNME	PF19086
TvSTS06	DEVSD	NDVYSYNME	PF19086
TvSTS07	DEQTD	NDLLSYRKE	PF19086; PF03936
TvSTS12	DDLSD	NDLCSFNKE	PF19086; PF03936
TvSTS14	DDLG**G**	NDILSFYKE	PF06330
TvSTS16	DDLP**G**	NDLLSFYKE	PF06330
IIIS	DEYTD	NDIASYNKE	PF19086
SiTPS	DDLMD	NDVYSFDNE	PF19086; PF03936
DS3	DDKLE	DLDTT	PF13419
LbSTS4a	DDITD	NDVYSYGKE	PF19086
LbSTS6	DDVF**Q**	NDVLSFYKE	PF06330

na: missing data; DDVF**Q**: amino acids replacements at the conserved motif active sites are underlined and in bold. DDFTDD: red color sequence indicates substitution of Motif II.

## Data Availability

Not applicable.

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
