# Peer review of "Molecular and Functional Analyses of Characterized Sesquiterpene Synthases in Mushroom-Forming Fungi"

_jof, 2023, doi:10.3390/jof9101017_

Round 1

Reviewer 1 Report

The authors make a comprehensive list of fungal sesquiterpene synthases previously reported in the literature. They then use this list as references to identify candidate sesquiterpene synthases in the basidiomycete Flammulina velutipes and experimentally characterize some of them.

Overall, the study appears to be well executed from a technical standpoint. The reporting could be improved in the following ways:

1. Include a supplementary table in comma-separated, tab-separated, or Excel format containing the data from tables 1 and 2 in the main text. This will make it much easier for other researchers to use the data.

2. It looks like some of the enzyme products were not compared to authentic standards, but only to the NIST17 library. These  compounds should not be called "Identified", they should be called "Putatively annotated". See section 2.9 of this paper: https://www.ncbi.nlm.nih.gov/pmc/articles/PMC3772505/ , or a summary here: https://www.vanderbilt.edu/cit/metabolite-identification-confidence-levels/

Author Response

Comments to the Author

The authors make a comprehensive list of fungal sesquiterpene synthases previously reported in the literature. They then use this list as references to identify candidate sesquiterpene synthases in the basidiomycete Flammulina velutipes and experimentally characterize some of them.

Overall, the study appears to be well executed from a technical standpoint. The reporting could be improved in the following ways.

Response: We appreciate your positive comments about the manuscript. We have carefully revised our manuscript according to your suggestions.

  1. Include a supplementary table in comma-separated, tab-separated, or Excel format containing the data from tables 1 and 2 in the main text. This will make it much easier for other researchers to use the data.

Response: Thank you very much for your comments and suggestions. We have added a supplementary table (Table S5) in excel format containing the data from tables 1 and 2 in new manuscript, according to your advice.

  1. It looks like some of the enzyme products were not compared to authentic standards, but only to the NIST17 library. These compounds should not be called "Identified", they should be called "Putatively annotated". See section 2.9 of this paper: https://www.ncbi.nlm.nih.gov/pmc/articles/PMC3772505/, or a summary here: https://www.vanderbilt.edu/cit/metabolite-identification-confidence-levels/.

Response: We appreciate the suggestions about the manuscript. We have corrected the relevant description in the corresponding places of the revised manuscript (new lines 254; 280).

Reviewer 2 Report

Extensive work has been done by the authors but some points are missing from the manuscript which are critical like detailed methodologies, aim of study, novelty points and conclusions are missing from the manuscript. Authors advised to revise the manuscript thoroughly before publication. The following points will be help for the revision-

1.      In introduction, author should clearly mention the aim of study and how this study is helpful to the reader. Furthermore, author should add few lines about novelty of study.

2.      Somewhere connections are missing to connect the phylogenetic and bioinformatics studies to the heterologous expression of sesquiterpenes.

3.      Table 1 should be a part of introduction or literature survey, this is not the research output of current study, so it can be shifted to other part or supplementary materials.

4.      Similarly, Figure 1 also should be part of introduction or review literature, author should shift this in another sections. Furthermore, the source (reference) for Figure 1 is not provided.

5.      In methodology, author should explain, how sesquiterpenes has been purified from S. cerevisiae strain Sc027? And how sample is prepared for GC-MS analysis.

6.      Section 2.5, author mention ‘codon optimization of FvSTS genes.’ Author should explain in detail, what is the need to optimize codons and how it has been proceeded.

7.      Author should give proper details for heterologous expression studies primer designing, primer sequence, PCR amplification conditions and transformation procedures, for this author can provide suitable references.  

8.      Section 2.6, GC-MS analysis, provide the details of some missing points inlet and detector temperature and detector type.  

9.      Conclusions is missing in the manuscript, provide the same at the end of manuscript.

10.  Some grammatical and punctuation errors have been notices, authors need to proofread. 

Moderate English check required 

Author Response

Comments to the Author

Extensive work has been done by the authors but some points are missing from the manuscript which are critical like detailed methodologies, aim of study, novelty points and conclusions are missing from the manuscript. Authors advised to revise the manuscript thoroughly before publication. The following points will be help for the revision.

Response: We appreciate the positive comments about the manuscript. We consider your suggestions are all significant for our research work. The criticisms and concerns have been adequately adopted and the revised manuscript has been modified according to your suggestions.

  1. In introduction, author should clearly mention the aim of study and how this study is helpful to the reader. Furthermore, author should add few lines about novelty of study.

Response: We are appreciated with your suggestions. We very much agree with your comments that it is necessary to perfect introduction section. We have added the corresponding description in the revised manuscript (new lines 66-76).

  1. Somewhere connections are missing to connect the phylogenetic and bioinformatics studies to the heterologous expression of sesquiterpenes.

Response: Thank you for this comment. First, we collated a database of characterized STSs in mushroom to analyze their catalytic products, conserved domains and motifs, and phylogenetic tree. Then, we have obtained FvSTS08 from Flammulina velutipes by bioinformatics approach and have experimentally characterized that FvSTS08 could synthesize β-barbatene. Finally, we used FvSTS08 to achieve heterologous high-yield β-barbatene production in S. cerevisiae. We have added the corresponding description in the revised manuscript (new lines 259-261).

  1. Table 1 should be a part of introduction or literature survey, this is not the research output of current study, so it can be shifted to other part or supplementary materials.

Response: Thank you for this comment. We have collected relevant information of mushroom STSs, then summarized, classified and made Table 1 by ourself. The information of clade in table 1 was also obtained and analyzed using MEGA7 software by ourself. So, we think Table 1 should not be shifted to another sections.

  1. Similarly, Figure 1 also should be part of introduction or review literature, author should shift this in another sections. Furthermore, the source (reference) for Figure 1 is not provided.

Response: Thanks for the constructive comments. The Figure 1 in this study is not the result of previous work. We have collated amino acid sequences of 172 mushroom STSs, then constructed the phylogenetic tree (Figure 1) by ourself using MEGA7 software. So, we think Figure 1 should not be shifted to another sections.

  1. In methodology, author should explain, how sesquiterpenes has been purified from S. cerevisiae strain Sc027? And how sample is prepared for GC-MS analysis.

Response: Thanks for your constructive comments. Sorry for the missing information. We have added the corresponding description in the revised manuscript (new lines 109-115; 440-441).

  1. Section 2.5, author mention ‘codon optimization of FvSTS genes.’ Author should explain in detail, what is the need to optimize codons and how it has been proceeded.

Response: Thanks for the constructive comments. Codon usage bias has a complex effect on protein expression levels when recombinant proteins are heterologously expressed. The codon optimization is the most critical determinant of increasing protein expression. In this study, three FvSTS genes were codon-optimized for expression in S. cerevisiae by GENEWIZ (Suzhou, China) and synthesized into pESC-URA vector. We have added the corresponding description in the revised manuscript (new lines 238-241; 513-514).

  1. Author should give proper details for heterologous expression studies primer designing, primer sequence, PCR amplification conditions and transformation procedures, for this author can provide suitable references.

Response: Thank you for this comment. We have added Figure S2 to explain strain construction of heterologous expression to guide primer design. For primer sequence, we have given proper detail information in Table S3. For PCR amplification conditions, we referred to the instructions of Phanta Max Super-Fidelity DNA Polymerase (Vazyme, P505) and have cited the enzyme in the revised manuscript (new lines 97-98). The lithium acetate method is a common and high-efficiency method for yeast transformation and has been widely used in many previous studies. We have can provided the reference in the revised manuscript (new lines 99; 436-437).

  1. Section 2.6, GC-MS analysis, provide the details of some missing points inlet and detector temperature and detector type.

Response: Thank you for this comment. We have provided the corresponding detail description in the revised manuscript (new lines 118; 122).

  1. Conclusions is missing in the manuscript, provide the same at the end of manuscript.

Response: Thanks for the constructive comments. We have added the conclusions section in the revised manuscript (new lines 343-352).

  1. Some grammatical and punctuation errors have been notices, authors need to proofread.

Response: Thanks for the constructive comments. The language in our revised manuscript has been professionally polished and verified by a native English-speaking editor at Editeg.

Round 2

Reviewer 2 Report

The sufficient corrections has been done in the manuscript.

Author should check one more time for minor English mistakes and punctuation error